# Effects of Elevated CO_2_ Concentration on Host Adaptability and Chlorantraniliprole Susceptibility in *Spodoptera frugiperda*

**DOI:** 10.3390/insects13111029

**Published:** 2022-11-07

**Authors:** Zhihui Lu, Zhongxiang Sun, Yahong Li, Ruoshi Hao, Yaping Chen, Bin Chen, Xiaoping Qin, Xuan Tao, Furong Gui

**Affiliations:** 1State Key Laboratory of Conservation and Utilization of Biological Resources of Yunnan, College of Plant Protection, Yunnan Agricultural University, Kunming 650201, China; 2Yunnan Plant Protection and Quarantine Station, Kunming 650034, China; 3Yunnan Plateau Characteristic Agriculture Industry Research Institute, Kunming 650201, China

**Keywords:** carbon dioxide, *Spodoptera frugiperda*, host adaptability, chlorantraniliprole, detoxification

## Abstract

**Simple Summary:**

Elevated carbon dioxide concentrations (eCO_2_) have a significant direct effect on herbivorous insects during their host seeking and oviposition. On the other hand, eCO_2_ could dramatically alter leaf chemistry of plants, especially in C3 plants (such as wheat), which in turn is likely to affect the population performance of insects that feed on the host plant. However, the effects of eCO_2_ on host adaptability and insecticide resistance in the fall armyworm, *Spodoptera frugiperda*, are unclear. In this study, we demonstrate that elevated CO_2_ concentrations increased the population performance of *S. frugiperda* on wheat and reduced the susceptibility of *S. frugiperda* to chlorantraniliprole by inducing the expression of detoxification enzyme genes. This report warns that *S. frugiperda* may continue to be a major global pest through better host adaptation and increased insecticide resistance in the future as atmospheric CO_2_ continues to rise.

**Abstract:**

Elevated atmospheric carbon dioxide concentrations (eCO_2_) can affect both herbivorous insects and their host plants. The fall armyworm (FAW), *Spodoptera frugiperda,* is a highly polyphagous agricultural pest that may attack more than 350 host plant species and has developed resistance to both conventional and novel-action insecticides. However, the effects of eCO_2_ on host adaptability and insecticide resistance of FAW are unclear. We hypothesized that eCO_2_ might affect insecticide resistance of FAW by affecting its host plants. To test this hypothesis, we investigated the effect of eCO_2_ on (1) FAW’s susceptibility to chlorantraniliprole after feeding on wheat, (2) FAW’s population performance traits (including the growth and reproduction), and (3) changes in gene expression in the FAW by transcriptome sequencing. The toxicity of chlorantraniliprole against the FAW under eCO_2_ (800 µL/L) stress showed that the LC_50_ values were 2.40, 2.06, and 1.46 times the values at the ambient CO_2_ concentration (400 µL/L, aCO_2_) for the three generations, respectively. Under eCO_2_, the life span of pupae and adults and the total number of generations were significantly shorter than the FAW under aCO_2_. Compared to the aCO_2_ treatment, the weights of the 3rd and 4th instar larvae and pupae of FAW under eCO_2_ were significantly heavier. Transcriptome sequencing results showed that more than 79 detoxification enzyme genes in FAW were upregulated under eCO_2_ treatment, including 40 P450, 5 CarE, 17 ABC, and 7 UGT genes. Our results showed that eCO_2_ increased the population performance of FAW on wheat and reduced its susceptibility to chlorantraniliprole by inducing the expression of detoxification enzyme genes. This study has important implications for assessing the damage of FAW in the future under the environment of increasing atmospheric CO_2_ concentration.

## 1. Introduction

Carbon dioxide (CO_2_) levels in the atmosphere have risen dramatically as a result of the Industrial Revolution. Today, the concentration of atmospheric CO_2_ is about 400 parts per million (www.esrl.noaa.gov/gmd/ccgg/trends/, accessed on 11 March 2022), but it could be twice as high by the end of the century [1].

High atmospheric CO_2_ levels could have a significant effect on insects. A number of hematophagous arthropods respond directly to the level of CO_2_ during host seeking and oviposition [2,3]. Research on *Frankliniella occidentalis* (Pergande) and *Frankliniella intonsa* (Trybom) have shown that elevated CO_2_ concentration (800 µL/L, eCO_2_) decreases development duration and increases the fecundity and daily eggs laid per female [4], implying more severe damage in the future. Nevertheless, most studies have focused only on the direct effects of eCO_2_ on insects. The amount of plant material consumed by phytophagous insects is inextricably linked to their suitability and nutritional quality. In addition to increasing growth rates, eCO_2_ in the atmosphere dramatically alters these plant traits, such as leaf chemistry [3]. The effect is especially noticeable for C_3_ plants, such as wheat [5]. Half of all insects, including most Orthoptera, Hemiptera, and Lepidoptera, feed on plants as larvae. Therefore, the amount of plant material taken up by insects is mainly determined by the nutrient composition of the plant. Among a variety of factors that influence plant nutritional quality, nitrogen is the most important [6,7]. In some cases, it is very common for the carbon/nitrogen (C:N) ratio in the leaf tissue to greatly increase, implying a reduction in food quality. According to the compensatory feeding theory, insects may need to eat more foliage to obtain enough nitrogen-based nutrition (mainly proteins). Therefore, phytophagous insects are indirectly affected by changes in their host plants [8]. The development, egg laying capacity, reproduction, and adult longevity of insect herbivores may be influenced by changes in host plant quality and quantity as a result of eCO_2_ [9].

The fall armyworm (FAW), *Spodoptera frugiperda* (*Lepidoptera: Noctuidae*), is an invasive species native to North America and is currently distributed in Asia [10], Africa, Oceania, Central America and South America (https://gd.eppo.int/taxon/LAPHFR/distribution, accessed on 11 March 2022). The FAW is regarded as a super pest because of its superior biological characteristics (including being highly polyphagous, and rapidly developing resistance to insecticides) [11,12,13]. In China, FAW causes serious damage to crop yield both directly or indirectly. The four provinces most affected by FAW in China are Yunnan ($830.51 M), Guangxi ($346.09 M), Sichuan ($116.87 M), and Shandong ($116.43 M) [14]. However, as a super-invasive pest, the impacts of eCO_2_ on the FAW in the future have not yet been predicted.

eCO_2_ could directly amplify the effect of insecticides on insects that feed only under eCO_2_ stress. Previous studies have shown that combining methyl bromide with an increase in 20% CO_2_ concentration increases the susceptibility of adult *Sitophilus oryzae* to methyl bromide by 1.5-fold [15]. eCO_2_ amplifies the efficacy of spinetoram on *Frankliniella occidentalis* and *F. intonsa* [16], which indicates that eCO_2_ increases the insecticidal activity of spinetoram against *F. occidentalis* and *F. intonsa* compared with aCO_2_. In the field, however, eCO_2_ tends to have a combination effect on herbivorous insects and host plants at the same time, rather than only having direct effects on insects. Therefore, it is important to study the comprehensive effects of eCO_2_ on insects and plants, as both insects and their host plants are exposed to eCO_2_ stress. Rao et al. found that in the *Spodoptera litura* Fab–peanut system, eCO_2_ caused a higher LC_50_ value of spinosad and deltamethrin with a comprehensive impact, showing a ‘reduction of toxicity’ [17]. In addition, the susceptibility of the brown plant hopper *Nilaparvata lugens* to triazophos was significantly decreased in eCO_2_ compared to aCO_2_ levels [18]. Thus, the efficacy of insecticides on insects at different CO_2_ concentrations depends on the experimental treatment. The direct and combined effects of CO_2_ may result in different insecticidal efficacies in pesticides. To date, the effect of eCO_2_ on the efficacy of insecticides against the FAW remains unclear. Previous studies have reported that elevated CO_2_ increases the activity of detoxifying enzymes such as carboxylesterases (CarEs) and glutathione S-transferases (GSTs) [16]. Detoxification enzymes play essential roles in the survival of insects exposed to adverse environments [19,20,21], therefore, higher detoxifying enzymes activity indicate that those enzymes may be involved in anti-eCO_2_ stress, thereby changing the insecticide susceptibility in insect. However, the exact mechanism is still unclear. Chlorantraniliprole is a diamide insecticide that has high efficacy against lepidopteran insects, low toxicity to mammals and beneficial insects, and absence of cross-resistance with traditional insecticides. Chlorantraniliprole has been shown to have a high effect on FAW control and is widely used to control the FAW in many commercial crops worldwide [22,23,24,25]. Therefore, it is important to determine the effect of eCO_2_ on the chlorantraniliprole susceptibility of the FAW.

eCO_2_ concentrations tend to have a combined effect on herbivorous insects and host plants, but most of the current studies only consider the effect in a direct or indirect way. The FAW is a devastating pest, and it is necessary to predict its potential damage in the future after the atmospheric CO_2_ concentration increases. To test the hypothesis that eCO_2_ might affect insecticide resistance of FAW by affecting its host plants, in this study, we first investigated how eCO_2_ affects the susceptibility of the FAW to chlorantraniliprole, then calculated the growth and reproduction of the FAW to analyse its population performance traits under eCO_2_. Finally, we used transcriptomic analysis and qRT-PCR to identify detoxification genes in the FAW induced under eCO_2_ stress.

## 2. Materials and Methods

### 2.1. Plant Materials and Insect Stocks

Wheat (Huai mai 36), *Triticum aestivum* L, is a C3 plant susceptible to eCO_2_, and is one of the FAW’s favourite host plants. Wheat was grown hydroponically in two separate climate chambers with two CO_2_ concentrations (400 and 800 µL/L) under the same temperature, light intensity, and humidity regimes at Yunnan Agricultural University, Kunming, Yunnan Province, China (25°07′ N, 102°44′ E). Wheat leaves under both treatment conditions were selected for FAW feeding when each plant was at least 8 days old. Wheat was watered every day, and no chemical fertilizer or insecticide was used throughout the experiment. The two treatments on wheat were named W_A_ (wheat that grew at aCO_2_) and W_E_ (wheat that grew at eCO_2_).

The tested FAWs were collected from Yuanjiang, Yunnan Province, China (23°35′59.52″ N, 101°58′39.64″ E) in May 2019. The larvae were reared on an artificial diet without exposure to any pesticides since then [26], and the adults were fed with a 10% honey/water solution in the laboratory under the conditions of 27 ± 0.5 °C, 70 ± 5% RH and a photoperiod of 16 h:8 h (L:D).

### 2.2. Effect of eCO_2_ on FAW Population Performance

To accurately record the effect of eCO_2_ on the FAW, two levels of atmospheric CO_2_ concentration, eCO_2_ (800 µL/L, the predicted level at the end of this century) and aCO_2_ (400 µL/L, the current atmospheric CO_2_ level), were set up in two artificial climate chambers (LTC-1000, SANTN, Shanghai, China), with 16 h light at 27 °C and 8 h dark at 25 °C, and 70% relative humidity (RH). CO_2_ gas was supplied to the climate chamber all day, and the CO_2_ concentrations were monitored and adjusted automatically once every 20 min. Eggs laid by the same female were separately placed into two climate chambers with different CO_2_ concentrations. Newly hatched larvae were randomly selected and reared individually in glass vials (d = 2.5 cm, covered with circular filter paper at the bottom of the vial). We set up three treatments: FAWs grown at 400 µL/L CO_2_ fed with wheat grown at aCO_2_ (named F_A_W_A_), FAWs grown at 800 µL/L CO_2_ fed with wheat grown at aCO_2_ (named F_E_W_A_), and FAWs grown at 800 µL/L CO_2_ fed with wheat grown at eCO_2_ (named F_E_W_E_). Each treatment had three replicates, and each replicate contained 25 individual larvae. Larval instar and body weight changes were recorded daily, all vials were cleaned, and leaves were replaced daily until the larvae pupated.

### 2.3. Effect of eCO_2_ on the Chlorantraniliprole Susceptibility of the FAW

The toxicity effect of chlorantraniliprole on the FAW under the three treatments mentioned above (F_A_W_A,_ F_E_W_A,_ and F_E_W_E_) were assessed and adapted from a previous study [16]. Chlorantraniliprole was diluted with distilled water to five concentrations. A larval rearing box (18 cm × 12 cm × 8 cm) and wheat leaves that grew under two carbon dioxide concentrations (elevated and ambient) were dipped for 2 h and 15 s in a chlorantraniliprole suspension and dried at room temperature. The control treatments were dipped in distilled water. Thirty random fourth larvae were selected and placed in three rearing boxes (each rearing box placed 10 larvae). The mortality rate of the FAWs was assessed after 48 h. The experiments were performed in triplicate. FAWs were presumed dead when they showed no reaction when touched with a brush. The concentration mortality regression equation and LC_50_ of chlorantraniliprole against the FAWs were derived and calculated. The above experiments were performed under eCO_2_ and aCO_2_. The same bioassay was performed in three generations of larvae.

### 2.4. Effect of eCO_2_ on Wheat

To investigate the influence of two CO_2_ concentrations on wheat (W_A_ and W_E_), the length of wheat shoots and roots were measured after eight days of cultivation, after which those shoots and roots were oven-dried at 80 °C for 72 h. The dry biomass of the roots and shoots was recorded to determine the ratio of roots to shoots, and the relative biomass of the roots and shoots. For each treatment, data on the weight of the total dry plant were used to represent their biomass.

### 2.5. RNA Isolation, Transcriptome Library Preparation and Sequencing

To understand how eCO_2_ affects FAW gene expression, comparative transcriptomic analyses were carried out on larvae F_A_W_A_ and F_E_W_E_. Larvae at the fifth instar were selected from three treatments. Five larvae were selected as one sample for the experiment, and three biological replicates of each concentration were performed. A total RNA extraction kit (RNeasy Mini Kit, Qiagen, Hilden, Germany) was used for RNA extraction. RNase-free agarose gel was used to check for contamination. RNA integrity and purity were measured using an Agilent 2100 Bioanalyzer system (Agilent, Santa Clara, CA, USA) and Nano Drop Spectrophotometer (THERMO, Waltham, MA, USA), respectively. The extracted RNA was reverse transcribed to cDNA for library preparation. The libraries were prepared following the manufacturer’s instructions using the BGISEQ-500 sequencing platform. Pair-end sequencing with 100 bp in length was performed using a BGISEQ-500 sequencer with the processed libraries.

The sequencing data were filtered with SOAPnuke (v1.5.2, Source: https://github.com/BGI-flexlab/SOAPnuke, accessed on 20 August 2021) by removing reads (1) containing a sequencing adapter, (2) whose low-quality base ratio (base quality less than or equal to five) was more than 20%, and (3) whose unknown base (‘N’ base) ratio was more than 5%; afterwards, clean reads were obtained and stored in FASTQ format [27]. The clean reads were mapped to the reference genome using HISAT2 (v2.0.4, Source: http://www.ccb.jhu.edu/software/hisat/index.shtml, accessed on 27 August 2021) [28]. The clean reads were aligned to the reference genome [12]. Bowtie2 (v2.2.5, Source: http://bowtiebio.sourceforge.net/%20Bowtie2%20/index.shtml, accessed on 28 August 2021) was applied to align the clean reads to the reference coding gene set, then expression level of gene was calculated by RSEM (v1.2.12, Source: https://github.com/deweylab/RSEM, accessed on 31 August 2021) [29,30]. The heatmap was drawn with pheatmap (v1.0.8, Source: https://cran.r-project.org/web/packages/pheatmap/index.html, accessed on 1 September 2021) according to the gene expression in different samples [31]. Differential expression analysis was performed using DESeq2 (v1.4.5, http://www.bioconductor.org/packages/release/bioc/html/DESeq2.html, accessed on 1 September 2021) with a *Q* value ≤ 0.05 [32]. Genes that were differentially expressed in each comparison group with |log2(fold change in a comparison group) > 1 and adjusted *p* value ≤ 0.001 were considered differentially expressed genes (DEGs). To compare the changes in phenotype, GO (http://www.geneontology.org/, accessed on 1 September 2021) and KEGG (https://www.kegg.jp/, accessed on 1 September 2021) enrichment analysis of annotated DEGs was performed using Phyper (https://en.wikipedia.org/wiki/Hypergeometric_distribution, accessed on 2 September 2021) based on the Hypergeometric test. The significance levels of terms and pathways were determined by a *Q* value with a rigorous threshold (*Q* value ≤ 0.05).

### 2.6. qRT-PCR

Two micrograms of total RNA of each sample were used for qRT-PCR cDNA synthesis using a FastKing RT Kit (with gDNase) (KR116, TIANGEN, Beijing, China). The TransStart^®^ Tip Green qPCR SuperMix (AQ141, TransGen Biotech, Beijing, China) was used for qRT-PCR in a 10 μL reaction solution on a LighCycler480 II machine (Roche, Basel, Switzerland). qRT-PCR proceeded as follows: one cycle of denaturation at 94 °C for 3 min, followed by 40 cycles of denaturation at 94 °C for 10 s, annealing at 60 °C for 10 s, and elongation at 72 °C for 20 s, followed by melting curve analysis. Two reference genes, RPL10-insects and RPL13-JIA, were selected for normalisation of the qRT-PCR results. mRNA levels were analysed using the 2^−△△CT^ method [33]. Each assay was repeated three times. Primers were designed using Primer 5.0 software. The primer sequences are listed in Appendix A.

### 2.7. Data Analyses

Each experiment was conducted with three biological replicates, and all data were expressed as the mean ± standard error (SE). The larvae weight and development period of the FAW, wheat shoot and root length, and dry biomass were analysed in a data analysis model based on an independent sample *t*-test (DMRT) (*p* < 0.05) in SPSS 24.0.0 (IBM, Armonk, NY, USA). Heatmaps were plotted using Origin Pro 2021b (64-bit) SR1 (9.8.5.204 Learning Edition). Bar graphs were plotted using GraphPad Prism version 8.0.0 for Windows (GraphPad Software, San Diego, CA, USA, www.graphpad.com, accessed on 15 March 2022) and Excel 2019.

## 3. Results

### 3.1. Effect of eCO_2_ on the Susceptibility of the FAW to Chlorantraniliprole

Log-probit regression analyses for the toxicity of different chlorantraniliprole concentrations against the FAW showed that the insecticidal activity of chlorantraniliprole was higher for F_A_W_A_ than F_E_W_E_ (Table 1). The LC_50_ values for F_E_W_E_ were 2.40, 2.06, and 1.46 times that for F_A_W_A_ (Table 1). The results showed that F_A_W_A_ was more susceptible to chlorantraniliprole than F_E_W_E_ in all three generations.

To clarify the direct effect of eCO_2_ on the FAW, the toxicity of chlorantraniliprole against F_E_W_A_ treatment was determined in the first generation (Appendix A), and the results showed that the insecticidal activity of chlorantraniliprole was 1.84 times higher for F_E_W_A_ than F_A_W_A_, indicating that F_E_W_A_ was more susceptible to chlorantraniliprole than F_A_W_A_.

### 3.2. Effect of eCO_2_ on FAW Population Performance in Wheat

To examine the effect of eCO_2_ on the FAW, F_A_W_A_ and F_E_W_E_ were reared. Compared with F_A_W_A_, the pupae, adult, and total generation durations of F_E_W_E_ were significantly shorter (Figure 1A). Compared with the F_A_W_A_, fourth instar larvae of F_E_W_E_ and the pupae were heavier (Figure 1B). The average female fecundity of F_A_W_A_ and F_E_W_E_ was 918.70 and 811.80, respectively (Figure 1C); though F_E_W_E_ had a lower fecundity, there was no significant difference.

### 3.3. Effect of eCO_2_ on Wheat Biomass

The biomass of wheat was affected by the CO_2_ level. Overall, the weight and length of wheat shoots grown at eCO_2_ were significantly higher than those grown at aCO_2_ (Figure 2A–C). The root weight and length of wheat nurtured at eCO_2_ were higher than those nurtured at aCO_2_. Wheat grown at a high CO_2_ concentration had a lower root-to-shoot ratio (Figure 2D).

### 3.4. Effect of eCO_2_ on the DEGs of FAW

To investigate the transcriptomic profiles of the FAW under different CO_2_ conditions, the treatment (F_A_W_A_) and control (F_E_W_E_) larvae groups were used for transcriptomic analyses (Appendix A). The expression of all identified genes obtained through RNA sequencing was compared for F_A_W_A_ and F_E_W_E_. There were 1542 DEGs between the two groups, with 897 upregulated and 645 downregulated genes in F_E_W_E_ (Figure 3). Among the upregulated genes, 78 genes belonged to detoxification enzyme-related genes; these DEGs may lead to changes in insect tolerance to insecticides.

### 3.5. GO and KEGG Analyses

DEGs in the 2 comparison groups were annotated using the Gene Ontology (GO) function database, which divided them into three macroscopic groups, namely: biological process, cellular component, and molecular function (Appendix A). For F_A_W_A_ versus F_E_W_E_, DEGs were assigned to 1594 GO terms enriched in 893 terms of biological process, 212 terms of cellular component, and 489 terms of molecular function, respectively (Figure 4). Among these enriched GO terms, many have growth and development- and detoxification and metabolism-related functions, including monooxygenase activity, oxidoreductase activity, carbohydrate transport, chitin binding, cholinesterase activity, response to stress, UDP-glucose 4-epimerase activity, and so on.

Unigenes with KEGG annotations were classified into five major categories (Appendix A). For the secondary categories, the pathways of carbohydrate metabolism, global and overview maps, and lipid metabolism were ranked as the top three subcategories in each category. There are a series of DEGs related to carbohydrate metabolism, energy metabolism, environmental adaptation, and lipid metabolism in categories. A total of 20 signalling pathways (Table 2) were enriched in F_A_W_A_ compared to F_E_W_E_, and a number of DEGs related to glutathione metabolism, fatty acid metabolism, ABC transporters and peroxisome were also enriched.

### 3.6. Detoxification Enzyme Gene Differentially Expression and Validation

Among the 897 upregulated DEGs, many detoxification genes were upregulated, including 40 genes encoding cytochrome P450 monooxygenases (P450s) (Figure 5A), 17 ATP-binding cassette transporters (ABCs) (Figure 5B), five carboxylesterases (CarEs) (Figure 5C), 7 UDP glucosyltransferases (UGTs) (Figure 5D), four acetylcholinesterase (AchEs) (Figure 5E), and five glutathione S-transferases (GSTs) (Figure 5F).

To confirm the results of the transcriptomic analyses, 16 detoxification enzyme genes were selected for qRT-PCR validation. The expression patterns of the selected 14 detoxification enzyme genes significantly upregulated expression in F_E_W_E_ based on qRT-PCR analysis (Figure 6). The changes in gene expression levels based on qRT-PCR were largely consistent with the transcriptomic data.

## 4. Discussion

In addition to the evolution of pest resistance to insecticides, conditional resistance can result from the reduction of insect’s susceptibility to insecticides under changed environmental conditions [34]. Rao et al. found that in *S. litura*, higher CO_2_ concentrations caused higher LC_50_ values for spinosad and deltamethrin, while it caused lower LC_50_ values for flubendiamide, emamectin benzoate, and quinalphos, indicating that the comprehensive influence of eCO_2_ on insect resistance depends on the pesticide type [17]. In this study, F_A_W_A_ was more susceptible to chlorantraniliprole than F_E_W_E_ in three generations (Table 1). Similarly, the susceptibility of *N. lugens* to triazophos was significantly decreased in eCO_2_ compared to aCO_2_ levels. Ge et al. suggested that this is because eCO_2_ accelerates the dissipation of triazophos in rice [18]. However, how pests resist pesticides at eCO_2_ is also important. Recently, transcriptomic analysis has become a routine method for identifying DEGs in insects in response to environmental stress [35,36]. We screened expanded gene families in the FAW to determine how eCO_2_ affects the susceptibility of the FAW to pesticides. We compared RNA transcription levels using RNA-seq. We determined the DEGs among FAW treated by F_A_W_A_ (FAW grown at 800 µL/L CO_2_ fed with W_A_) and F_E_W_E_ (FAW grown at 400 µL/L CO_2_ fed with W_E_). Our results show that there were 1542 DEGs, with 897 upregulated and 645 downregulated genes in F_A_W_A_ compared to F_E_W_E_ (Figure 3), and that a large number of DEGs were involved in metabolic detoxification, including P450s, ABCs, CarEs, UGTs, and GSTs (Figure 5). This indicates that eCO_2_ can induce the expression of detoxification enzyme genes, which is most likely caused by changes in the concentration of chemical substances in wheat leaves.

Some studies have found that eCO_2_ can amplify the effect of chemicals on insects. For instance, eCO_2_ inhibits respiratory enzymes, such as malic enzymes and succinate dehydrogenase, resulting in decreased adenosine triphosphate (ATP) generation. Insects may die as a result of insufficient energy supply. eCO_2_ levels may increase membrane permeability, allowing more insecticides to enter the insect’s body [37,38,39,40]. Compared to aCO_2_, high atmospheric CO_2_ can directly amplify the effect of spinetoram insecticidal activity against Thysanoptera pest *F. occidentalis* and *F. intonsa* [16]. In this study, when FAW was only directly affected by eCO_2_ (F_E_W_A_), it was more susceptible to chlorantraniliprole than F_A_W_A_ (Appendix A); this conclusion is consistent with Fan et al. [16].

Previous studies suggest that the CO_2_ increase may affect plants, as it alters the chemical composition of the air, leading to modifications in plants’ secondary metabolism. Increases in the C:N ratio have been seen in plants growing at high CO_2_ concentrations, which are expected to affect carbon-based secondary chemistry. As a result of these changes, plant tissue nutritional quality is reduced, resulting in an increase in phenolics and a decrease in nitrogen in the plants [8,41,42,43,44]. Under elevated CO_2_, the reduction in N concentration across a broad range of species can exceed 14%, with C3 plants responding more than C4 plants [3]. To find out how eCO_2_ affects insecticide resistance of FAW by affecting its host plants, wheat (one of the FAW’s favourite C3 plants) biomass was measured. In this study, wheat biomass at two CO_2_ concentrations suggests its tissue nutritional quality, C:N ratio, secondary metabolism and defence chemistry content were changed (Figure 2) [3,45,46]. Similarly, many studies have shown that eCO_2_ increases secondary metabolism and defence chemistry content in plants. A review by Robinson reported that plants grown in eCO_2_ environments increased total phenolics and condensed tannins and flavonoids by 19%, 22%, and 27%, respectively [8]. eCO_2_ increased the concentration of quercetin, kaempferol, and fisetin in leaves and rhizomes of two ginger varieties, and exhibited more enhanced free radical scavenging power [47]. In soybean (*Glycine max*), quercetin-to-kaempferol ratios increase as a result of a strong increase in aliphatic glucosinolates and the methylsulfinylalkyl glucosinolates glucoraphanin and glucoiberin [48]. The total glucosinolate content increased in broccoli (*Brassica oleracea*) and *Arabidopsis thaliana* cultivated at eCO_2_ [49,50]. When insects feed on plants with altered secondary metabolism and defence chemistry, the expression of detoxification enzyme genes will change, which will affect their susceptibility to insecticides. Lu et al. found that in *Spodoptera litura*, pre-exposure to flavone induced detoxification gene expression and effectively increased larval tolerance to multiple synthetic insecticides [51]. After dietary exposure to xanthotoxin, the 20E signalling pathway and detoxification enzyme genes were modulated by the ROS/CncC pathway to improve tolerance of *Spodoptera litura* larvae to λ-cyhalothrin [52]. The activities of *Spodoptera litura* P450 and *CYP6AB60* transcription levels were significantly elevated after exposure to an artificial diet supplemented with the plant secondary metabolites coumarin, xanthotoxin, or tomatine [53]. Therefore, we speculated that eCO_2_ not only changes the growth, development, and reproduction of FAW, but also enhanced biodegradation of xenobiotics by overproduction of a complex array of detoxification enzymes, such as cytochrome P450 monooxygenases (P450s), carboxy/cholinesterases (CCEs), ATP-binding cassette transporter (ABCs), and glutathione S-transferases (GSTs) [54,55,56,57]. Therefore, eCO_2_ increases wheat’s secondary metabolism and defence chemistry content to induce FAW detoxification enzyme gene upregulation, thereby decreasing F_E_W_E_ susceptibility to chlorantraniliprole.

Insect’s growth, fecundity, occurrence, and population distribution could change with environmental stress as a result of metabolic rate fluctuation [4]. Elevated atmospheric CO_2_ concentration may have effects on insects directly or indirectly [3]. The oviposition period, sex ratio, net reproductive rate, intrinsic rate of increase, and finite rate of increase of *F. occidentalis* increased under eCO_2_ conditions, while larval duration, survival rate, mean generation time, and population doubling time decreased [58]. For the cotton bollworm, *Helicoverpa armigera* (Hubner), the direct effects of eCO_2_ significantly increased mortality and decreased fecundity [59]. When the Asian corn borer, *Ostrinia furnacalis* (Guenee), was fed an artificial diet under eCO_2_ conditions, it had a longer larval and pupal development time and decreased rates of survival and mean relative growth [60]. In terms of indirect effects, Qian et al. found that after feeding on plants grown under eCO_2_, acetylcholinesterase, carboxylesterase, and mixed-function oxidase activity in thrips increased to counter plant defences. Greater thrip densities induced stronger plant defences and, in turn, detoxifying enzyme levels in thrips increased [19]. In this study, the larval, pupae, adult, and total generation duration of the FAW between F_A_W_A_ and F_E_W_E_ were significantly different; larvae and pupae weight were also different, but average female fecundity between the two treatments was not significantly different (Figure 1), indicating that eCO_2_ results in faster population outbreak and more serious damage (heavier larvae and pupae) with no change in fecundity of FAW. In the future, the damage caused to crops by the FAW may increase due to a shorter developmental duration, a heavier body weight, and no difference in fertility.

## 5. Conclusions

In conclusion, we found that eCO_2_ could upregulate many detoxification genes in the FAW *Spodoptera frugiperda*, which were likely to be involved in insecticide susceptibility of FAW at eCO_2_. Furthermore, eCO_2_ increased the population performance of the FAW on host plants. These two responses of the FAW to eCO_2_ may further cause a FAW population outbreak and increase the damage caused by the FAW in the future.

## Figures and Tables

**Figure 1 insects-13-01029-f001:**
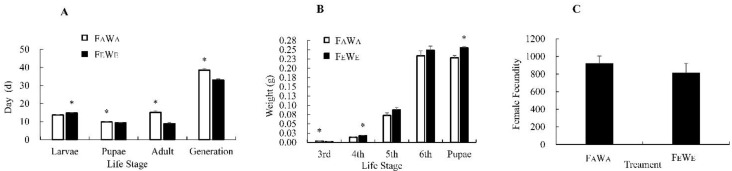
Development period (**A**); larvae weight (**B**); female fecundity (**C**) (Mean ± SE) of F_A_W_A_ and F_E_W_E_. Note: Asterisks denote a significant difference between ambient and eCO_2_ by the independent-sample *t*-test at *p* < 0.05.

**Figure 2 insects-13-01029-f002:**
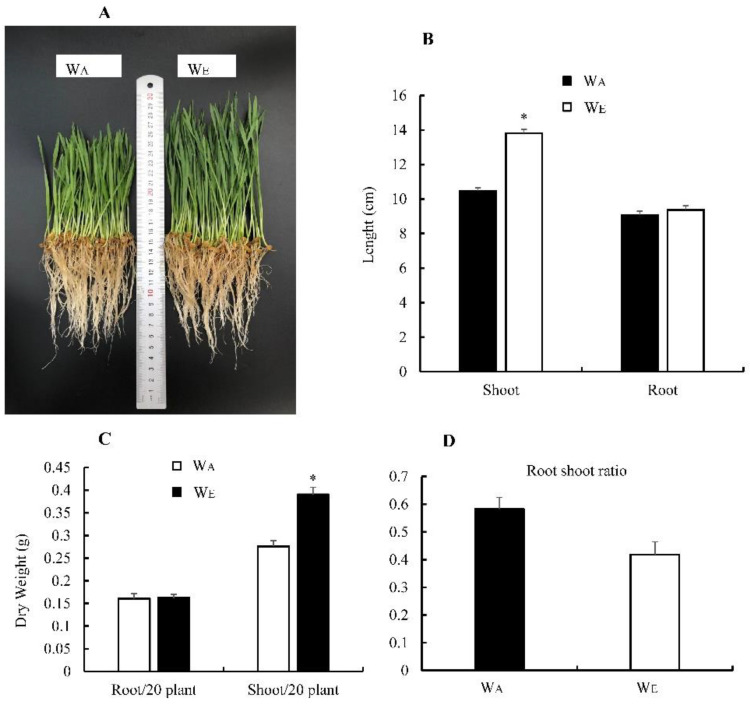
Length (**A**,**B**); root and shoot weight (**C**); and root-to-shoot ratio (**D**) of wheat at aCO_2_ and eCO_2_. Note: Asterisks denote significant difference between ambient and elevated CO_2_ by the independent-sample *t*-test at *p* < 0.05. W_A_: wheat that grew at aCO_2_, W_E_: wheat that grew at eCO_2_. The same is true for the following figures.

**Figure 3 insects-13-01029-f003:**
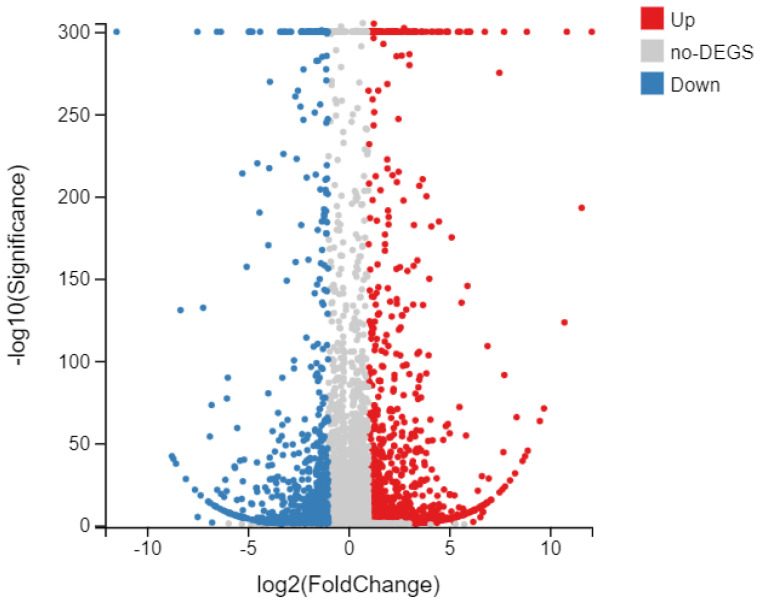
DEGs in F_A_W_A_ versus F_E_W_E_. A volcano plot of DEGs obtained from different comparative analyses. DEGs upregulated in the FAW are represented by red dots, and those that were downregulated are represented by blue dots; genes with no significant differences in expression are represented in grey.

**Figure 4 insects-13-01029-f004:**
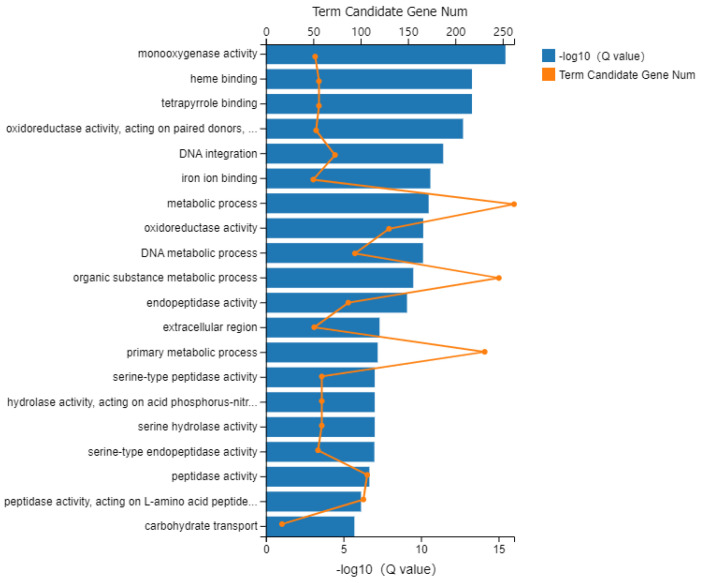
GO function enrichment analysis. GO enrichment results for DEGs in F_A_W_A_ versus F_E_W_E_. Note: The length of the X-axis column represents the size of the *Q* value (−log10(*Q* value)), and the value of the point on the polyline on the upper X is the number of differential genes annotated to the GO term.

**Figure 5 insects-13-01029-f005:**
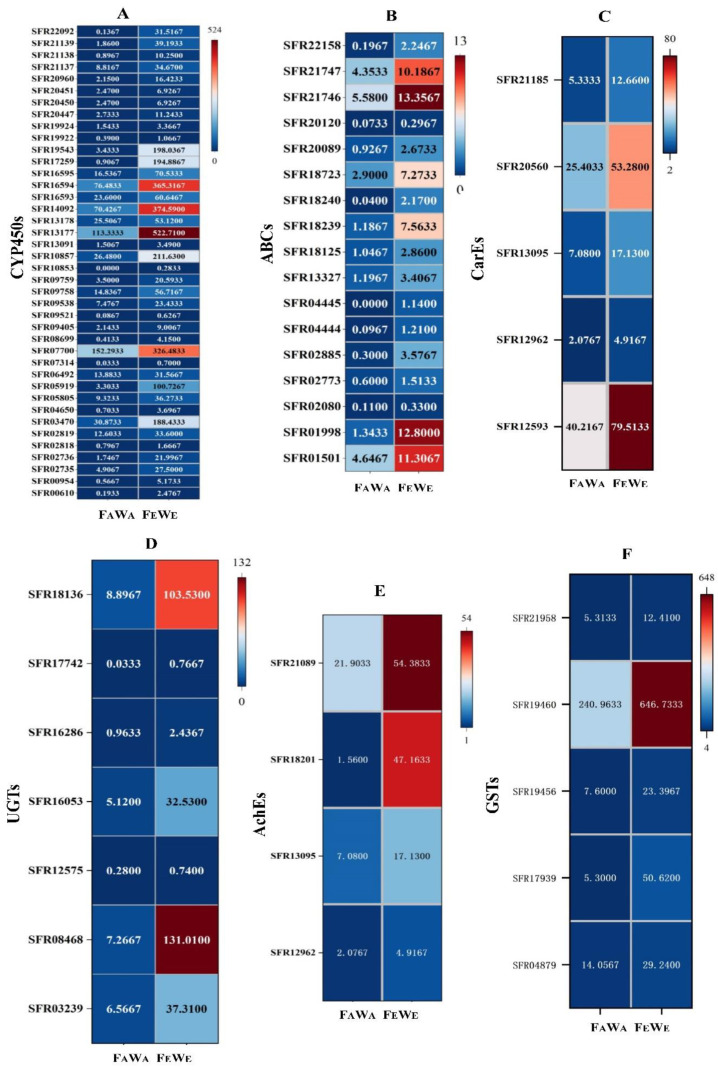
Heatmap of differentially expressed genes of F_A_W_A_ versus F_E_W_E_. (**A**) Heatmap of cytochrome P450 monooxygenase (P450) DEGs. (**B**) Heatmap of ATP-binding cassette transporter (ABC) DEGs. (**C**) Heatmap of carboxylesterase (CarE) DEGs. (**D**) Heatmap of UDP glucosyltransferase (UGT) DEGs. (**E**) Heatmap of acetylcholinesterase (AchE) DEGs. (**F**) Heatmap of glutathione S-transferase (GST) DEGs. Note: Heatmaps show the average values of FPKM. The x-axis shows the different treatments (F_A_W_A_ and F_E_W_E_). The colour represents the fold change of DEGs; red indicates upregulation, and blue indicates downregulation.

**Figure 6 insects-13-01029-f006:**
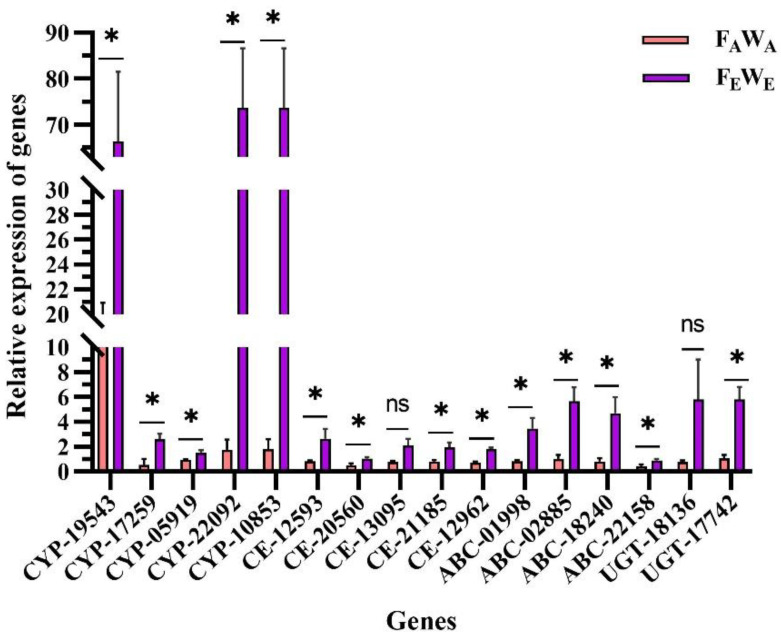
Quantitative real-time PCR (qRT-PCR) data of selected genes. Sixteen upregulated DEGs were selected for PCR analysis. RPL10-insects and RPL13-JIA were used as reference genes for qRT-PCR normalisation. The mRNA expression levels of the selected genes were calculated using the 2^−△△CT^ method. Note: Asterisks denote significant difference between ambient and elevated CO_2_ by the independent-sample *t*-test at *p* < 0.05, ns indicates insignificant.

**Table 1 insects-13-01029-t001:** Toxicity effect of chlorantraniliprole on 4th larvae of the FAW under eCO_2_ and aCO_2_.

Generation	Treatment	Concentration Response Regression Equation	*χ2*	*p*	LC_50_ (mg L^−1^) 95% CI
1st	F_A_W_A_	y = 4.2826 + 0.7447x	51.97	0.0001	9.19 (1.57−59.67)
	F_E_W_E_	y = 2.6994 + 1.7117x	33.90	0.0001	22.08 (9.00−64.63)
2nd	F_A_W_A_	y = 3.4718 + 1.5021x	40.16	0.0001	10.41 (4.25−27.20)
	F_E_W_E_	y = 3.9791 + 0.7667x	8.71	0.0334	21.45 (13.49−39.63)
3rd	F_A_W_A_	y = 4.2871 + 0.5890x	23.20	0.0001	16.82 (13.44−51.72)
	F_E_W_E_	y = 3.5639 + 1.0336x	7.99	0.0463	24.52 (12.96−71.98)

Note: 95% CI, 95% confident intervals; LC_50_, the concentration of chlorantraniliprole that is lethal to 50% of FAWs. The fall armyworm grown at 400 µL/L CO_2_ fed with wheat grown at ambient CO_2_ concentration (400 µL/L, aCO_2_) was named F_A_W_A_, and FAWs grown at 800 µL/L CO_2_ fed with wheat grown at elevated atmospheric carbon dioxide concentrations (800 µL/L, eCO_2_) were named F_E_W_E_, similarly hereinafter.

**Table 2 insects-13-01029-t002:** KEGG pathway enrichment results of DEGs in F_A_W_A_ versus F_E_W_E_.

Pathway ID	Pathway Name	*p* Value	*Q* Value
ko00563	Glycosylphosphatidylinositol (GPI)-anchor biosynthesis	0.000000001	0.0000000581
ko00965	Betalain biosynthesis	0.000000001	0.0000000581
ko00950	Isoquinoline alkaloid biosynthesis	0.000000016	0.0000006040
ko00790	Folate biosynthesis	0.000015700	0.0004409770
ko00350	Tyrosine metabolism	0.000021100	0.0004722640
ko03450	Non-homologous end-joining	0.000074600	0.0013931500
ko03420	Nucleotide excision repair	0.000169718	0.0027154880
ko00310	Lysine degradation	0.000531000	0.0074340010
ko00052	Galactose metabolism	0.000914600	0.0113816900
ko00590	Arachidonic acid metabolism	0.001703890	0.0173487000
ko00561	Glycerolipid metabolism	0.002804281	0.0261732900
ko03060	Protein export	0.005070793	0.0436868300

Note: Enriched signalling pathways with *Q <* 0.05 were considered statistically significant.

## Data Availability

The raw sequencing data from this study have been deposited in the Genome Sequence Archive in BIG Data Centre (https://ngdc.cncb.ac.cn), Beijing Institute of Genomics (BIG), Chinese Academy of Sciences, under the accession number: PRJCA012875.

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
