# Peer review of "Effects of Elevated CO2 Concentration on Host Adaptability and Chlorantraniliprole Susceptibility in Spodoptera frugiperda"

_insects, 2022, doi:10.3390/insects13111029_

Round 1

Reviewer 1 Report

The manuscript "Effects of Elevated CO2 Concentration on Host Adaptability of Spodoptera frugiperda and its Susceptibility to Chlorantraniliprole” revealed the effects of CO2 on FAW insecticide resistance using RNA-seq and toxicity assay, and the authors found that eCO2 induced the expression of more than 79 detoxification enzyme genes in FAW, increased FAW resistance to chlorantraniliprole, and eCO2 also increased the population performance of FAW on wheat. It’s a great idea to analyze the CO2 change effects on insect development and detoxification activity due to the increasing CO2 emissions. However, there are some problems to be addressed in the current version before it can be accepted for publication.

1.       In the RNA-seq samples, the treatments are FaWa and FeWe, so, the treatment has two factors: eCO2 and eCO2-wheat leaves, based on your conclusions, it seems likely that eCO2 is the main factor to change the gene expression in FAW, while it’s hard to say there are no effects of eCO2 wheat leaves. Please reconsider the design, should you set up a FaWe control to clarify the effects? Based on the result line 244, I think the eCO2 wheat leaves may be the main factor.

2.       I found a new publication that reported the P450 CYP4BQ1 expression is under the control of ROS/CncC signaling pathway, and in your RNA-seq data, there are a lot of related genes that are upregulated, so can you pull out some genes related to ROS/CncC pathway to check their relationship? Whether the eCO2 enhanced the expression of detoxification genes via upregulated ROS? Did you ever measure the ROS changes after your treatment?

3.       Line 242, Fig1C showed that there is no fecundity difference between FaWa and FeWe, but your conclusion is that eCO2 increased the FAW population on wheat, do you think there is a conflict?

4.       The detoxification enzyme background in the introduction is missing.

5.       Figures 1 and 2 are not high resolution.

6.       Font size in line 379-384 and line 392 is bigger than others.

7.       Section 3.3 showed the eCO2 promoted the growth of wheat and biomass, please add some sentences to connect with why you use the eCO2 wheat leaves to feed FAW for the RNA-seq sample collection, there is a gap for readers.

8.       There are a lot of RNA-seq analyses these years, but the data from the companies is just data, they are just numbers, so when you show these numbers, you should give them a biological life to enrich your manuscript, not just the number. You can conclude each analysis in your result section to make sure they are alive and try to explain why the number is changed in your treatments, which will be helpful for readers and improve the quality of the manuscript.

9.      In Figure 5, not all the genes are upregulated in FeWe based on the color scale, while in section 3.6 you said they were upregulated. Please correct them.

10.    Did you deposit the RNA-seq data in database?

Author Response

Response to Reviewer 1 Comments

Please see revised manuscript in the attachment

The manuscript "Effects of Elevated CO2 Concentration on Host Adaptability of Spodoptera frugiperda and its Susceptibility to Chlorantraniliprole” revealed the effects of CO2 on FAW insecticide resistance using RNA-seq and toxicity assay, and the authors found that eCO2 induced the expression of more than 79 detoxification enzyme genes in FAW, increased FAW resistance to chlorantraniliprole, and eCO2 also increased the population performance of FAW on wheat. It’s a great idea to analyze the CO2 change effects on insect development and detoxification activity due to the increasing CO2 emissions. However, there are some problems to be addressed in the current version before it can be accepted for publication.

  1. In the RNA-seq samples, the treatments are FaWa and FeWe, so, the treatment has two factors: eCO2 and eCO2-wheat leaves, based on your conclusions, it seems likely that eCO2 is the main factor to change the gene expression in FAW, while it’s hard to say there are no effects of eCO2 wheat leaves. Please reconsider the design, should you set up a FaWe control to clarify the effects? Based on the result line 244, I think the eCO2 wheat leaves may be the main factor.

Response 1: Thank you very much for your approval and kindly comments, and    your comments are valuable and constructive, which are very helpful for us to revise and improve our manuscript. In our experiment, we set FAWA and FEWE treatments for three main reasons: 1. eCO2 concentrations tend to have a combined effect on herbivorous insects and host plants, but most of the current studies only consider the effect on a direct or indirect way. 2. If carbon dioxide concentration increases in the atmospheric, the insect-host plant system in the field will be under the same conditions together, therefore, it is more practical and meaningful to study the comprehensive effects of eCO2 concentration on insects. 3. We referred to some literatures with similar experimental designs to ours, such as Kanle Satishchandra et al (2018) and Rao et al (2021). Kanle Satishchandra et al evaluated the effects of different CO2 concentrations (380 and 550 ppm) on the life table of Tuta absoluta at two CO2 concentrations. Rao et al investigated the influence of two dimensions of climate change i.e., elevated temperature (eTemp) and elevated CO2 (eCO2) on toxicity of insecticides against Spodoptera litura Fab. on peanut.

       However, the suggestions you mentioned were also taken into consideration, we are agreeing and appreciate your suggestion. Set up a FaWe does help clarify whether eCO2 affects insect susceptibility to insecticides primarily through direct effects on insects or indirect effects on host plants. We have discussed in detail how eCO2 increases secondary metabolism and defense chemistry content to induce FAW detoxification enzyme gene upregulation, thereby decreasing FEWE susceptibility to chlorantraniliprole in paragraph 3 of the Discussion section of the revised manuscript. In the further study, we will fully consider your constructive suggestions to explore this issue in more depth.

Reference:

Kanle Satishchandra N, Vaddi S, Naik S O, et al. Effect of temperature and CO2 on population growth of South American tomato moth, Tuta absoluta (Meyrick) (Lepidoptera: Gelechiidae) on tomato[J]. Journal of economic entomology, 2018, 111(4): 1614-1624.

Rao M S, Sreelakshmi P, Deekshita K, et al. Interactive effects of temperature and CO2 on efficacy of insecticides against Spodoptera litura Fab. in a global warming context[J]. Phytoparasitica, 2021, 49(3): 417-431.

  1. I found a new publication that reported the P450 CYP4BQ1 expression is under the control of ROS/CncC signaling pathway, and in your RNA-seq data, there are a lot of related genes that are upregulated, so can you pull out some genes related to ROS/CncC pathway to check their relationship? Whether the eCO2 enhanced the expression of detoxification genes via upregulated ROS? Did you ever measure the ROS changes after your treatment?

Response 2: Thanks for your professional comments. We have carefully read the references you have mentioned. The study about (+)-α-pinene induces CYP4BQ1 via activation of the ROS/CncC signaling pathway in Dendroctonus armandi is very interesting and constructive, we have listed it as references in Line 104 of our revised manuscript. In our experiment, we mainly focus on the key phenomena of eCO2 in enhancing host adaptability and chlorantraniliprole susceptibility in Spodoptera frugiperda, and the possible mechanism was preliminarily studied. The suggestion you put forward is very helpful for us to further study the molecular mechanism behind this phenomenon. It is well known that both transcription factors and signaling pathways (such as basic leucine zipper (bZIP), basic-helix-loop-helix/ Per-ARNT-Sim and nuclear receptors superfamilies) are involved in the regulation of P450 genes in xenobiotic responses. Elevated CO2 concentrations regulate P450 genes through which transcription factor or/and signaling pathway requires a lot of effort to investigate. In the present study, we have found some differentially expressed genes, and in the further study, we will refere some literatures (for example, Dai et al 2021, Liu et al 2022) to select some genes such as CYP-19543, CYP-22092, CYP-10853 etc. to study the relationship between transcription factors/ signaling pathway and those genes.

References:

Dai, L.; Gao, H.; Chen, H. Expression Levels of Detoxification Enzyme Genes from Dendroctonus armandi (Coleoptera: Curculionidae) Fed on a Solid Diet Containing Pine Phloem and Terpenoids. Insects, 2021, 12, 926. https://doi.org/10.3390/insects12100926.

Liu B.; Tang M.; Chen H. Activation of the ROS/CncC Signaling Pathway Regulates Cytochrome P450 CYP4BQ1 Responsible for (+)-α-Pinene Tolerance in Dendroctonus armandi. Int. J. Mol. Sci. 2022, 23: 11578. https://doi.org/10.3390/ijms231911578.

  1. Line 242, Fig1C showed that there is no fecundity difference between FaWa and FeWe, but your conclusion is that eCO2 increased the FAW population on wheat, do you think there is a conflict?

Responses 3: We are sincerely appreciated your professional review and thank you very much for helping us to point out this issue. We didn't express the issue clearly before. Although eCO2 had no significant effect on fecundity, it significantly shortened the developmental duration and increased the larval weight, which would enable FAW to reach a larger population and harm host crops in a shorter time. We have updated this expression in the revised manuscript with “In this study, the larval, pupae, adult and total generation duration of the FAW between FAWA and FEWE were significantly different, larvae and pupae weight were also different too, but average female fecundity between the two treatments has no significantly different (Fig. 1), indicating that eCO2 result in faster population outbreak, more serious damage (heavier larvae and pupae) with no change in fecundity of FAW. In the future, the damage caused to crops by the FAW may increase due to a shorter developmental duration, a heavier body weight and no difference in fecundity.” in line 416-422.

  1. The detoxification enzyme background in the introduction is missing.

Response 4: The background about detoxification enzyme has been added in the introduction line101-107 of the revised manuscript.

  1. Figures 1 and 2 are not high resolution.

Response 5: We have changed Figure 1 and 2 with high resolution pictures in line 258 and 269.

  1. Font size in line 379-384 and line 392 is bigger than others.

Response 6: Thank you very much, the font size in line 379-384 and line 392 has been adjusted to the same with others.

  1. Section 3.3 showed the eCO2 promoted the growth of wheat and biomass, please add some sentences to connect with why you use the eCO2 wheat leaves to feed FAW for the RNA-seq sample collection, there is a gap for readers.

Response 7: Thanks a lot for raise this issue. Under elevated CO2, the reduction in N concentration across a broad range of species can exceed 14%, with C3 plants responding more than C4 plants (Zavala et al., 2013). To find out how eCO2 affect insecticide resistance of FAW by affecting its host plants, wheat (one of the FAW’s favorite C3 plant) biomass was measured. Therefore, wheat is the most suitable model plant for this experiment. That’s why we use the eCO2 wheat leaves to feed FAW for the RNA-seq sample collection. The relevant descriptions and literature have been supplemented in the revised manuscript (Lines 369-373).

Reference:

Zavala J A, Nabity P D, DeLucia E H. An emerging understanding of mechanisms governing insect herbivory under elevated CO2[J]. Annual review of entomology, 2013, 58(1): 79-97.

  1. There are a lot of RNA-seq analyses these years, but the data from the companies is just data, they are just numbers, so when you show these numbers, you should give them a biological life to enrich your manuscript, not just the number. You can conclude each analysis in your result section to make sure they are alive and try to explain why the number is changed in your treatments, which will be helpful for readers and improve the quality of the manuscript.

Responses 8: Thank you for your professional comments. After re-checking the RNA-seq analyses data, we have added a series of descriptions about GO and KEGG analyses in the revised manuscript in line 292-295 and line 304-308.

  1. In Figure 5, not all the genes are upregulated in FeWe based on the color scale, while in section 3.6 you said they were upregulated. Please correct them.

Responses 9: We are apologizing for the mistake of the previous manuscript. All the genes are upregulated in FeWe, we have added a new graph for figure 5. For section 3.6, we repeated the qRT-PCR for each cDNA synthesis, increased the number of experimental replicates and redrew the graphs. Finally, we found that 14 of 16 upregulated DEGs were significantly up-regulated expression in FEWE based on re-qRT-PCR analysis. The statement for Fig. 6 has also been changed from “The expression patterns of the selected detoxification enzyme genes were significantly up-regulated expression in FEWE based on qRT-PCR analysis (Fig. 6). The changes in gene expression levels based on qRT-PCR were largely consistent with the transcriptomic data.” to “The expression patterns of the 14 selected detoxification enzyme genes were significantly up-regulated expression in FEWE based on qRT-PCR analysis (Fig. 6). The changes in gene expression levels based on qRT-PCR were largely consistent with the transcriptomic data.”

  1. Did you deposit the RNA-seq data in database?

Responses 10: Yes, the raw sequencing data of the present study have been deposited in the Genome Sequence Archive in BIG Data Center (https://ngdc.cncb.ac.cn), Beijing Institute of Genomics (BIG), Chinese Academy of Sciences, under the accession number: PRJCA012875.

Reviewer 2 Report

The authors investigated the effects of CO2 on the biological characteristics of Spodoptera frugiperda and their host plant, wheat. Besides, comparative transcriptomic analyses were performed to reveal the mechanism of the reduced chlorantraniliprole susceptibility of S. frugiperda affected by CO2. The methods used in this manuscript are scientifically sound and the results are also clearly presented. However, there were some questions listed as follows:

1.     The resolution of the images in the paper needs to be adjusted to meet the requirements of journal publication.

2.     In section 2.2 and 2.3, the authors mentioned a population FEWA, while the data concerning this population should also be shown in the manuscript. The discussion of these results could also be supplemented.

3.     The authors used the fourth instar larvae for the chlorantraniliprole susceptibility experiments, why was the fifth instar larvae selected for the RNA-seq? Under normal circumstances, the larval stage for these experiments should be consistent.

4.     In Figure 6, only seven DEGs showed similar results in RT-qPCR and RNA-seq. More DEGs should be selected for DEG vilification.

Author Response

Response to Reviewer 2 Comments

Please see revised manuscript in the attachment

The authors investigated the effects of CO2 on the biological characteristics of Spodoptera frugiperda and their host plant, wheat. Besides, comparative transcriptomic analyses were performed to reveal the mechanism of the reduced chlorantraniliprole susceptibility of S. frugiperda affected by CO2. The methods used in this manuscript are scientifically sound and the results are also clearly presented. However, there were some questions listed as follows:

  1. The resolution of the images in the paper needs to be adjusted to meet the requirements of journal publication.

Response 1: Thanks for your careful checks. We have changed all Figure with high resolution pictures in manuscript.

  1. In section 2.2 and 2.3, the authors mentioned a population FEWA, while the data concerning this population should also be shown in the manuscript. The discussion of these results could also be supplemented.

Response 2: We are sincerely appreciated your professional review and thank you very much for pointing out the problems to us. The reason of in section 2.2 and 2.3, our mentioned a population FEWA, while the data concerning this population not fully shown in the manuscript is: If carbon dioxide concentration increases in the atmospheric, the insect-host system in the field will be under the same conditions together, therefore, it is more practical and meaningful to study the comprehensive effects of eCO2 concentration on insects. That's why we only set FAWA and FEWE treatments for RNA-seq, and the experimental design was referenced to Kanle Satishchandra et al and Rao et al. (Kanle Satishchandra et al evaluated the effects of different CO2 concentrations (380 and 550 ppm) on the life table of Tuta absoluta at two CO2 concentrations. Rao et al investigated the influence of two dimensions of climate change i.e., elevated temperature (eTemp) and elevated CO2 (eCO2) on toxicity of insecticides against Spodoptera litura Fab. on peanut). The reason for adding the FEWA treatment is that some studies shown that compared to aCO2, high atmospheric CO2 can directly amplify the effect of spinetoram insecticidal activity against Thysanoptera pest F. occidentalis and F. intonsa (Fan et al., 2022). We wanted to verify whether our results were consistent with previous studies. In this study, we found that when FAW was only directly affected by eCO2 (FEWA), it was more susceptible to chlorantraniliprole than FAWA, this conclusion is consistent with others. And the discussion about this part has been supplemented at Line 355-363 of the revised manuscript.

Reference:

Kanle Satishchandra N, Vaddi S, Naik S O, et al. Effect of temperature and CO2 on population growth of South American tomato moth, Tuta absoluta (Meyrick) (Lepidoptera: Gelechiidae) on tomato[J]. Journal of economic entomology, 2018, 111(4): 1614-1624.

Rao M S, Sreelakshmi P, Deekshita K, et al. Interactive effects of temperature and CO2 on efficacy of insecticides against Spodoptera litura Fab. in a global warming context[J]. Phytoparasitica, 2021, 49(3): 417-431.

Fan Z, Qian L, Chen Y, et al. Effects of elevated CO2 on activities of protective and detoxifying enzymes in Frankliniella occidentalis and F. intonsa under spinetoram stress[J]. Pest Management Science, 2022, 78(1): 274-286.

  1. The authors used the fourth instar larvae for the chlorantraniliprole susceptibility experiments, why was the fifth instar larvae selected for the RNA-seq? Under normal circumstances, the larval stage for these experiments should be consistent.

Responses 3: Good question. In our experiment, we used 4th instar 1-day-old larvae for chlorantraniliprole susceptibility experiments, and after 48 hours of treatment with pesticides, most of the larvae that survived were 5th instar. Therefore, we collected newly 5th instar larvae for the RNA-seq to consistent the treatment.

  1. In Figure 6, only seven DEGs showed similar results in RT-qPCR and RNA-seq. More DEGs should be selected for DEG vilification.

Responses 4: Thanks for your kindly suggestion. According to you suggest, we repeated the qRT-PCR for each cDNA synthesis, and increased the number of experimental replicates and redrew the graphs for Fig 6. Finally, we found that 14 of 16 upregulated DEGs were significantly up-regulated expression in FEWE based on re-qRT-PCR analysis.

Reviewer 3 Report

This is an excellent study with clear and convincing data. The authors do an especially good job in placing their findings in the context of other research in the area. I found no substantial problems with the manuscript and have only a few suggested corrections in language, as indicated below:

l. 17 change "alters" to "alter"
l. 20 change "remained" to "are"
l. 28 remove "types"
l 29 change "remained" to "are"
l 32 insert "and" before "(3)
l 33 lower case "c"  (Changes to change)
l 51 "expected to be" to "could be" (let's hope it doesn't reach 800ppm!
l 62-64 feed on plants as larvae
l 319 change "due to the fact that" to "because"
l 321 change "are" to "is"

Author Response

Response to Reviewer 3 Comments

Please see revised manuscript in the attachment. 

This is an excellent study with clear and convincing data. The authors do an especially good job in placing their findings in the context of other research in the area. I found no substantial problems with the manuscript and have only a few suggested corrections in language, as indicated below:

  1. 17 change "alters" to "alter"

Responses 1: Thanks for your kindly suggestion. The error has been revised in line 19 of the revised manuscript.

  1. 20 change "remained" to "are"

Responses 2: Has been revised in line 22.

  1. 28 remove "types"

Responses 3: Has been removed in line 30.

l 29 change "remained" to "are"

Responses 4: Has been revised in line 31.

l 32 insert "and" before "(3)

Responses 5: Has been revised in line 34.

l 33 lower case "c" (Changes to change)

Responses 6: Has been revised in line 35.

l 51 "expected to be" to "could be" (let's hope it doesn't reach 800ppm!)

Responses 7: Thanks for your careful checks. Has been revised in line 53.

l 62-64 feed on plants as larvae

Responses 8: Has been revised in line 66.

l 319 change "due to the fact that" to "because"

Responses 9: Has been revised in line 342.

l 321 change "are" to "is"

Responses 10: Has been revised in line 344.

Reviewer 4 Report

This manuscript reports investigation results about effects of elevated CO2 on host plant adaptability and insecticide susceptiblity in Spodoptera frugiperda. Under the trend of climatic change including increasing CO2, such study is of great significance for us to understand the responses of organisms including insect pests and crop plants and thereby to implement proper measures in pest managment. 

Experiments in this manuscript were reasonably  designed and conducted, and the results were basically presented properly. However, authors should make some improvements  before the manuscript could be accepted for publication.

(1) Title: I suggest to adjust the title to: Effects of Elevated CO2 Concentration on Host Adaptability  and Chlorantraniliprole Susceptibility in Spodoptera frugiperda.

(2) Line 128, some information of the artificial diet used to rear the larvae should be provided, such as composition, or references.

(3) Figures and Tables should be self-explained, i.e., some explanations or full spellings should be available in Notes of tables or figures for abbrevations or other necessary terms, such as FAW, aCO2, eCO2, FAWA, FAWE, etc., so that readers can understand most of the meanings wihtout reading the text.

(4) Please check words or grammatic usage, e.g., Line 16, its (their); Line 396, insecticide(s). Also, Line 169, FAW RNA isolation

Author Response

Response to Reviewer 4 Comments

Please see revised manuscript in the attachment.

This manuscript reports investigation results about effects of elevated CO2 on host plant adaptability and insecticide susceptibility in Spodoptera frugiperda. Under the trend of climatic change including increasing CO2, such study is of great significance for us to understand the responses of organisms including insect pests and crop plants and thereby to implement proper measures in pest management.

Experiments in this manuscript were reasonably designed and conducted, and the results were basically presented properly. However, authors should make some improvements before the manuscript could be accepted for publication.

  • Title: I suggest to adjust the title to: Effects of Elevated CO2 Concentration on Host Adaptability and Chlorantraniliprole Susceptibility in Spodoptera frugiperda.

Responses 1: We are sincerely appreciated your professional review and thank you very much for help us. We have adjusted the title.

  • Line 128, some information of the artificial diet used to rear the larvae should be provided, such as composition, or references.

Responses 2: Thanks for your comments and we are sorry for unclear statement. We have added reference to making artificial diet in Line135 of the revised manuscript.

  • Figures and Tables should be self-explained, i.e., some explanations or full spellings should be available in Notes of tables or figures for abbreviations or other necessary terms, such as FAW, aCO2, eCO2, FAWA, FAWE, etc., so that readers can understand most of the meanings without reading the text.

Response 3: Thank you very much for pointing out the issue to us. The explanations and full spellings have been added to the revised manuscript in Line238-241, 270-272.

(4) Please check words or grammatic usage, e.g., Line 16, its (their); Line 396, insecticide(s). Also, Line 169, FAW RNA isolation

Response 4: Thanks for your careful checks. We feel sorry for our carelessness. The mistake has been revised in line 18, 425 and 180 of the revised manuscript.

Round 2

Reviewer 1 Report

Line 94, Nilaparvata lugens should be italic, please double-check all the similar typos.

Author Response

Line 94, Nilaparvata lugens should be italic, please double-check all the similar typos.

Responses : We feel sorry for our carelessness.  We rechecked  all the Latin names in the manuscript and changed the formatting to italics.